# OpenReview forum: "BPO: Revisiting Preference Modeling in Direct Preference Optimization"
_ICLR.cc/2026/Conference — ICLR 2026 Conference Withdrawn Submission_

### Official Review · Reviewer_2Cgr · 2025-10-30

**Soundness:** 1
**Presentation:** 2
**Contribution:** 2
**Rating:** 2
**Confidence:** 4

**Summary:**

One common phenomenon with DPO is the decreasing likelihood of preferred responses and this work aims to address this. The authors propose using a balanced reward margin which takes the minimum of the reward of the preferred response and the negative reward of the unpreferred response. They also add a hyperparameter that weights the different rewards to control the extent to which the chosen response likelihood is increased. They demonstrate improved performance on math reasoning benchmarks compared to DPO and variants.

**Strengths:**

Novel Objective - The paper introduces a new objective to address the decrease in likelihood for preferred responses through the balanced reward margin which takes the minimum of the reward of the preferred response and the negative reward of the unpreferred response. This creates an optimization landscape that avoids likelihoods of both responses decreasing addressing the issue and they demonstrate improved performance on math reasoning benchmarks by reducing likelihood displacement.

Empirical results - The paper demonstrates that mitigating likelihood displacement improves performance on math reasoning benchmarks across a variety of datasets, base models, and loss types.

**Weaknesses:**

Experimental Setup - The experiments focus on applying DPO and variants to math reasoning, but math reasoning training is often done with RL methods such as PPO or GRPO and works that do utilize DPO involve editing responses/updating preference data such as the Llama 3 paper cited. DPO is also most widely applied to human preference data such as HH-RLHF or UltraFeedback, so it is unclear whether BPO would lead to improvements for more common applications of DPO. An evaluation of the methods on such preference data would make the contributions stronger.

Technical Details- The work claims that BPO leads to accelerated convergence and reduced computational overhead. However, there is not strong support for either of these claims. It is unclear based on the reward dynamics shown or the analysis that there is faster convergence in practice, especially as in Figure 7, multiple of the DPO curves appear to have rewards plateau before the corresponding BPO curves. Regarding computational overhead, since a comparison of the rewards for the two responses is needed to compute the loss and gradient, the computational graph and activations for both responses need to be stored. Due to this and the fact that the selection of the min does not affect the memory needed to compute a backward pass, the computational cost should not significantly change. Parts describing the key advantages of BPO should be revised or provide concrete support for these claims.

Related Works - The work should discuss [1] which theoretically analyzes the decrease in response likelihoods and how embedding structure influences this.

[1] Razin, Noam, et al. "Unintentional unalignment: Likelihood displacement in direct preference optimization." arXiv preprint arXiv:2410.08847 (2024).

**Questions:**

- Can you provide clarification on the accelerated convergence and reduced computational overhead?
- Can you provide results on how BPO behaves on common preference data settings, such as instruction-following?

---

### Official Review · Reviewer_Rned · 2025-10-31

**Soundness:** 3
**Presentation:** 3
**Contribution:** 2
**Rating:** 4
**Confidence:** 4

**Summary:**

In this paper, authors proposes a simple one-line modification to DPO using $min(r_w, - \alpha r_l)$ instead of their difference to balance chosen/rejected response optimization. This is a quick extension to the existing DPO algorithms that uses the Bradley Terry model. Authors claim that this solves the "degraded chosen responses" problem in mathematical reasoning tasks. However, the paper suffers from missing SOTA baselines (SimPO, BDPO, ORPO etc), narrow evaluation scope (math-only) among other things.

**Strengths:**

Strengths:
1. The paper is very well written
2. The paper methodology is indeed very simple and easy to integrate with DPO type algorithms resulting in adoptions
3. The authors provides a good set of experimental evaluation to begin with (some major comments on that look at weaknesses)
4. I like that the authors provide ablations with different loss types and tried two families of models

**Weaknesses:**

Weaknesses:
1. The field has moved way beyond DPO. The paper therefore lacks comparisons with key strong baselines, like SimPO or KTO or ODPO or BDPO or ORPO, some of which tackles the diminishing log prob of chosen samples as a problem. Only Vanilla DPO is not a reasonable baseline, obviously this is going to perform better than DPO. The authors must compare performance with SOTA right now that constitutes improvement over the main DPO algorithm and not restrict themselves to just the algorithms that are in this space. While I commend that the authors compare BPO with Cal-DPO and DPOP, I strongly feel this algorithm needs to be compared to the other SOTA to show the efficacy. Note: I am not asking the authors to do comparison with GRPO/GSPO type algorithms, but only with the direct preference type algorithms.

2. Results lack confidence intervals and significance testing, making improvement claims difficult to validate.Please add them and mention the seeds.

3. The authors claim that the BPO has superior capacity to navigate complex reasoning landscapes which is a strawman argument. It is unclear why, there is no intuition in the paper. At its base DPO type algorithms even for reasoning tasks would need to have a chosen vs a rejected reasoning trace perhaps during training, so it is not intuitive how this has superior performance compared to other pseudo RL algorithms like GRPO/GSPO equivalents.

4. Further it seems performance of BPO seems sensitive to the variation of $\alpha$ and loss as well. Its a bit difficult to understand what value to choose, its interplay with beta and the learning rate etc.

5. The authors have only provided benchmarks on mathematical tasks for some reason. It is unclear what this simple change does to the model’s performance on other types of tasks.

6. Why do the authors think the AIME24 score went down between Qwen 2.5 math 7b instruction model compared to base for BPO? That is highly counterintuitive.

7. Can the authors provide any intuitions to whether this algorithm is trivially extendible to the online or iterative DPO paradigm?

**Questions:**

see weaknesses.

---

### Official Review · Reviewer_9Evt · 2025-11-01

**Soundness:** 2
**Presentation:** 3
**Contribution:** 2
**Rating:** 4
**Confidence:** 4

**Summary:**

The paper identifies the issue Degraded Chosen Responses (DCR) of DPO, *i.e.* decreasing likelihood of chosen responses and increasing the risk of generating out-of-distribution responses. The authors introduce Balanced Preference Optimization (BPO), a novel framework that dynamically balances the optimization of chosen and rejected responses through two key components: balanced reward margin and gap adaptor. Experimental results on multiple mathematical reasoning tasks show the superiority of BPO.

**Strengths:**

- The paper provides a theoretical justification for BPO, including a gradient analysis and Theorem 1, which ensures the learned policy maintains a minimum likelihood for chosen responses and prevents the DCR problem.
- The paper is well-structured and clearly written. This clear presentation makes the paper's contributions easy to follow.

**Weaknesses:**

- The experimental evaluation is limited to mathematical reasoning tasks. Consequently, it remains unclear whether BPO can generalize to other prevalent alignment objectives, such as instruction-following, helpfulness, or harmlessness.
- While the paper claims "Accelerated Convergence" and "Reduced Computational Overhead" as key advantages of BPO over DPO, these claims are not supported by corresponding empirical evidence.

**Questions:**

- Could the authors provide comparative results for BPO on widely used LLM benchmarks such as HumanEval, MMLU, and IFEval?
- Could the authors evaluate BPO using alternative base models beyond Qwen2.5-Math-7B-Base to show the robustness of the method?

---

### Note · Authors · 2025-11-28

**Comment:**

After careful consideration and in light of the reviewers' constructive feedback, we have decided to withdraw our submission to incorporate major revisions. We sincerely appreciate the time and effort of the reviewers and area chairs.

**Withdrawal Confirmation:**

I have read and agree with the venue's withdrawal policy on behalf of myself and my co-authors.